# Controlling *Legionella pneumophila* in Showerheads: Combination of Remedial Intervention and Preventative Flushing

**DOI:** 10.3390/microorganisms11061361

**Published:** 2023-05-23

**Authors:** Marianne Grimard-Conea, Michèle Prévost

**Affiliations:** Industrial Chair in Drinking Water, Department of Civil, Mining and Geological Engineering, Polytechnique Montreal, Montreal, QC H3C 3A7, Canada; michele.prevost@polymtl.ca

**Keywords:** *Legionella pneumophila*, building plumbing, flushing, chlorination, stagnation

## Abstract

Shock chlorination and remedial flushing are suggested to address *Legionella pneumophila* (*Lp*) contamination in buildings or during their (re)commissioning. However, data on general microbial measurements (adenosine tri-phosphate [ATP], total cell counts [TCC]), and the abundance of *Lp* are lacking to support their temporary implementation with variable water demands. In this study, the weekly short-term (3-week) impact of shock chlorination (20–25 mg/L free chlorine, 16 h) or remedial flushing (5-min flush) combined with distinct flushing regimes (daily, weekly, stagnant) was investigated in duplicates of showerheads in two shower systems. Results showed that the combination of stagnation and shock chlorination prompted biomass regrowth, with ATP and TCC in the first draws reaching large regrowth factors of 4.31–7.07-fold and 3.51–5.68-fold, respectively, from baseline values. Contrastingly, remedial flushing followed by stagnation generally resulted in complete or larger regrowth in *Lp* culturability and gene copies (gc). Irrespective of the intervention, daily flushed showerheads resulted in significantly (*p* < 0.05) lower ATP and TCC, as well as lower *Lp* concentrations than weekly flushes, in general. Nonetheless, *Lp* persisted at concentrations ranging from 11 to 223 as the most probable number per liter (MPN/L) and in the same order of magnitude (10^3^–10^4^ gc/L) than baseline values after remedial flushing, despite daily/weekly flushing, unlike shock chlorination which suppressed *Lp* culturability (down 3-log) for two weeks and gene copies by 1-log. This study provides insights on the most optimal short-term combination of remedial and preventative strategies that can be considered pending the implementation of suitable engineering controls or building-wide treatment.

## 1. Introduction

*Legionella pneumophila* is an opportunistic pathogen that can be distributed in potable water systems and is transmitted through the inhalation of contaminated water aerosols, causing either Pontiac fever, a milder form of the infection, or Legionnaires’ disease, a severe pneumonia-like infection in vulnerable or immunocompromised individuals [1]. In the last decades, *Legionella* infections, and more specifically Legionnaires’ disease, were associated with an increasing health burden, reflected in a large number of hospitalizations and deaths [2]. 

Water safety plans (WSP) rely on a multi-barrier approach including the implementation of control strategies and environmental monitoring to manage risks associated with water in building plumbing [3]. Mostly in response to positive *Legionella* water samples or in the aftermath of nosocomial Legionnaires’ disease cases, corrective actions are performed to suppress its growth. Among emergency measures, shock chlorination with free chlorine concentrations exceeding maximum allowable levels for drinking water (10–50 ppm) over a more or less prolonged contact time period (1–24 h) is commonplace [4,5,6,7,8], and even more so when thermal shock conditions (65–75 °C) cannot be applied [9]. In building plumbing, several studies have nevertheless reported the long-term persistence of *Legionella* bacteria despite the implementation of repeated shock chlorination as a remedial treatment [10,11,12]. Recolonization or long-term recalcitrant *Legionella* positivity have been mostly attributed to (1) the limited penetration rate of chlorine into pipe biofilms [13,14], (2) the intracellular protection of *Legionella* within protozoan hosts [15,16,17,18], (3) the intrinsic resistance of *Legionella* to chlorine [19], and (4) the difficulty to reach target-free chlorine concentrations at all distal outlets in large, legacy, and complex buildings [20,21]. 

WSP and *Legionella* regulation guidelines further advocate for preventative (routine) flushing of water points that are irregularly used, although, if even mentioned, frequencies, duration, and flow conditions vary greatly. Some prescribe a weekly flush of taps [5,22,23,24,25,26,27,28,29], whereas others advise for more frequent flushes such as flushing on a daily basis [25,30] or twice a week to prevent water stagnation [8,26]. Nonetheless, there is yet to find a consensus on the most suitable flushing regime that prevents or controls the growth of *Legionella*. In one hospital building plumbing system, Gavaldà and colleagues demonstrated that the probability of recovering culturable *Legionella* in taps with occasional water use was multiplied by more than two-fold as compared to taps used on a daily basis [31]. However, the statistical analysis seemed rather based on qualitative observations, nor was there any indication on the amount of water necessary to account for daily or occasional water usage. In another hospital setting, Totaro and colleagues observed a dramatic reduction in *L. pneumophila* serogroups 2–14 in semi-flushed hot water samples after chlorine dioxide residuals were maintained (0–0.3 mg/L) by sectorial preventative flushing. A high flushing frequency of one minute every two hours at all five dead-end branches of the hospital water network was required to be effective, while flushing every six hours was not [32]. Similarly, *Legionella pneumophila* concentrations were found to be lowered by 6.3-fold in high water use taps (21 flushes/week) in a pilot-scale study as opposed to low water use taps (one flush/week) [33]. Such higher water use frequencies are, however, impractical in the context of routine manual flushing of taps, requiring instead the use of more expensive auto-flush devices. 

Similar strategies were also put forward by local jurisdictions worldwide to reduce microbial risks associated with stagnant water during COVID-19 pandemic building shutdowns. Flushing to renew aged water with fresh water as a one-time event (recommissioning) or with repeated routine flushing was frequently suggested in guidance documents [34,35,36,37,38], but showed only temporary benefits [39,40,41,42]. Moreover, shock chlorination was recommended if (1) occupants vulnerable to waterborne diseases were returning to the building [36,43,44,45,46], (2) a depressurization incident occurred during the closure [34], (3) the hot water system was turned off for energy conservation purposes during the closure [47], or (4) there was a strong suspicion or testing confirmation of microbial contamination (e.g., to *Legionella*) [43]. The European Society of Clinical Microbiology and Infectious Diseases (ESCMID) study group for *Legionella* infection further recommended shock chlorination with 50 ppm of free chlorine prior to building shutdown if water systems were to be drained, or if they had not recently been disinfected or experienced operational issues with temperature and disinfectant residuals [47].

Additionally, water flushing and shock chlorination of building plumbing are practices commonly recommended in the American Society of Heating, Refrigerating and Air-conditioning Engineers (ASHRAE) Standard 188 during the commissioning of new building plumbing [48]. If building occupancy is postponed between two and four weeks following shock chlorination, ASHRAE then requires a thorough flush of all water fixtures. If the delay exceeds four weeks or more, the need for shock chlorination, flushing, or both shock chlorination and flushing of unoccupied sectors should be reevaluated. In the province of Quebec (Canada), small seasonal building systems (e.g., managed by national parks) typically undergo flushing and shock chlorination before seasonal reopening [49], as per required as well by the Revised Total Coliform Rule in the United States [50]. However, it remains unclear how remedial or preventative flushing as well as shock disinfection procedures should be applied in the context of existing buildings undergoing construction, renovation, and demolition activities, which are fairly recurrent in large and older building plumbing. Although not every construction activity involves building plumbing, they can still lead to the closure of sectors of the building for a short to a prolonged period of time, followed by the gradual incoming of occupants. More specifically, Scanlon and colleagues highlighted the lack of prevention strategies including specific practices to tackle water management before, during, and after the completion of such activities to address the high prevalence of nosocomial waterborne infections associated to these activities for patients with overnight stays [51].

Studies with controlled variable water use patterns mimicking different water demand following either (re)commissioning flushing or shock chlorination are lacking to study their short-term efficacy towards the inactivation of *Legionella* and broader microbial indicators. Therefore, the main aim of this study was to assess the short-term (three weeks) effectiveness of the combination of remedial flushing or shock chlorination with preventative flushing (daily, weekly, left stagnant) on the occurrence of planktonic *L. pneumophila* and other microbial measurements in shower systems. It was hypothesized that such combinations of remedial and preventative actions can only temporarily limit the regrowth of *L. pneumophila* and that daily flushes are more beneficial at maintaining lower microbial loads. 

## 2. Materials and Methods

### 2.1. Shower System Description

Two large, grouped shower systems in which a single bimetallic strip thermostatic mixing valve (TMV) feeds 20 to 22 showerheads at once with mitigated water were selected. Shower systems and building plumbing were described in a previous study [40]. Both shower systems were left completely unused for 12 weeks (distal stagnation) prior to the start of this study despite the reopening of the building from July to September 2020. The building was shut down during the second COVID-19 pandemic lockdown (October 2020 to March 2021). As the study took place in November and December 2020, the building plumbing was under low water demand throughout the study (less than 5% of normal building occupancy). 

All investigated showerheads were manually activated to ensure a flush of 5 min by activating a pressure button with resulting flow duration per activation ranging from 10 s to more than one minute. Copper was identified as the piping material across the building plumbing including the main piping of the two large, grouped shower systems, except for the connecting pipe between each showerhead and timer valve, which was made of flexible polymer. The interior of each TMV casing, where cold and hot water are mixed, was made of several materials including brass, ethylene propylene diene monomer (EPDM), stainless steel, and other plastics (WATTS Eurotherm Ultramix^®^, North Andover, MA, USA). 

### 2.2. Study Timeline and Water Sample Collection

The first sampling event took place on 16 November 2020, after a 12-week period without any shower usage, after which shock chlorination with free chlorine and remedial flushing were immediately carried out on the same day. Then, sampling events were conducted in the following three weeks from each mitigation intervention, thereby on 23 November, 30 November, and 7 December 2020 (on Mondays) (Appendix A).

During each sampling event, the cold and hot water supplied to each TMV were first individually sampled (one liter) after a brief five-minute flush to assess the upstream water quality in the building plumbing. Afterwards, the first draw (one liter) of water from the interior of each TMV casing and their immediate connecting pipe filled with mitigated water was sampled. Then, in a subset of six showerheads per shower system, first draws (one liter) were collected, followed by five-min flush samples (one liter). A sterile plastic bag was used to collect water from each showerhead in order to facilitate water collection and reduce exposure to water aerosols. All water samples were collected in sterile high-density polyethylene (HDPE) bottles. Finally, on every following weekday (Monday to Friday), showerheads that were flushed on a daily basis (Section 2.2.2) were also assessed for free and total chlorine by collecting first draws and five-min flush samples (Appendix A). The incoming cold water from the municipal distribution system and the hot water return loop were sampled once (30 November 2020) after brief flushes of 5–15 min. 

In mid-September 2021, building managers improved thermal control in the hot water distribution system by removing (1) the mixing point between a fraction of the hot water return loop coming back to the water heaters and the hot water delivered to the building, which increased temperature in both lines by up to 5 °C, as well as (2) the fraction of the hot water recirculation loop that was pre-heated in a small plate heat exchanger before being supplied to the first heater of the series of four. These modifications were selected based on previous observations made during the characterization of the centralized hot water system [40]. Then, from mid-September 2021 to early January 2022, building managers implemented daily flushes (Mondays to Sundays) in selected shower systems of the building (7 shower systems covering about 90 showerheads out of 114) by (1) flushing for 15–30 min with mitigated water the rear-end showerhead of large grouped shower systems followed by a 30 s flush of each upstream showerhead, and (2) flushing for five minutes with mitigated water all showerheads of small grouped shower systems or independent ones (single shower). One last sampling campaign was carried out on 3 November 2021 at a subset of 27 showerheads to sample only culturable *L. pneumophila* at first draw. The building was closed to the public in accordance with new COVID-19 lockdown orders from 20 December 2021 to mid-March 2022, hence after the last sampling. 

#### 2.2.1. Remedial Interventions

Shock chlorination was performed with a diluted (50% *v*:*v*) solution of liquid sodium hypochlorite (commercial household bleach) which was injected into the shower system just above the TMV casing in the mitigated water pipe through an available sampling port. The solution was pumped at approximately 200–250 mL/min with the use of a Masterflex^®^ L/S^®^ digital standard drive pump (Cole-Parmer Instrument Company, Vernon Hills, IL, USA). Starting with the showerhead closer to the TMV and moving towards the rear-end one, each showerhead was flushed (5–40 min) until target-free chlorine concentrations of 20–25 mg/L were reached. Overall, free chlorine concentrations ranging from 21.9 to 25.2 mg/L were measured at the start of the contact time period of 16 h. At the end of the contact time period, free and total chlorine concentrations ranged from 0.31 to 1.68 mg/L and 0.67 to 2.15 mg/L, respectively, corresponding to minimum disinfection CTs of 298–1613 mg.min/L (concentration of free chlorine X contact time). The resulting large free chlorine demand (93–99%) can be attributed to the extended contact time applied, as well as the demand exerted from copper piping, deposits, and biofilm. Then, each showerhead was flushed for five minutes, corresponding overall to approximatively three water turnovers of the shower system plumbing, with mitigated water to restore chlorine levels back to normal values (free chlorine: 0.02–0.11 mg/L; total chlorine: 0.14–0.15 mg/L).

Remedial flushing was carried out as previously described [40]. In short, each showerhead was flushed for five minutes with mitigated water, starting with the showerhead closest to the TMV and moving towards the rear-end one. Altogether, roughly 1000 L of mitigated water were flushed during remedial flushing, which was equivalent to more than three complete water turnovers of the shower system plumbing.

#### 2.2.2. Controlled Preventative Flushing in Duplicates of Showerheads

Following both remedial interventions, a controlled preventative flushing strategy was implemented in both shower systems at a subset of six showerheads per shower system. Briefly, the two showerheads closest to each TMV were flushed on a daily basis from Monday to Friday for five minutes with mitigated water (25–40 °C). Then, the two showerheads located at the middle of each shower system were flushed weekly (on Mondays) with mitigated water (35–38 °C), whereas the last two rear-end showerheads were left stagnant for the remaining part of the study (Figure 1).

#### 2.2.3. Temperature Monitoring in the Shower Systems

Water temperature was monitored at several locations throughout the study, including the hot water leaving the heaters and the hot water return loop, the cold and hot water supplied to each shower system’s TMV, as well as the mitigated water leaving each TMV. Temperature dataloggers (OM-CP-TC Temp X Series, 4 channels, Omega, Saint-Eustache, QC, Canada) were directly attached to pipe segments without thermal insulation. Temperature was also recorded with small thermocouple data loggers (OM-EL-USB-TC-LCD, Omega, Saint-Eustache, QC, Canada) that were installed at a subset of the investigated showerheads including one showerhead per duplicates of each flushing strategy (daily, weekly, left stagnant) in both shower systems (Figure 1). Temperature monitoring was put in place to examine thermal regimes and ensure that controlled flushing frequencies were applied accurately during the study. 

### 2.3. Water Samples Processing

All water samples were immediately analyzed for onsite physico-chemical parameters, whereas laboratory measurements including intracellular adenosine tri-phosphate (ATP), flow cytometry, culturable *L. pneumophila* and water sample filtration, were processed within 12 h of sampling. 

#### 2.3.1. Onsite Environmental Measurements

Temperature was measured using a digital thermometer (−50–300 °C), while pH (0–14 pH unit), conductivity (0.01–200 mS/cm), and dissolved oxygen (0–20 mg/L) were assessed with the HQ40d™ portable meter (HACH, London, ON, Canada) whose probes were inserted in a beaker containing approximatively 150–200 mL of the well-mixed water sample. Two successive aliquots of 10 mL were withdrawn from each well-mixed one-liter sample for assessment of free and total chlorine concentrations (0 to 2.00 mg/L), based on the HACH DPD Powder Pillows methods 8021 and 8167, respectively, with the portable DR 2800TM spectrophotometer (HACH, London, ON, Canada). During shock chlorination, dilutions (1:15) were made to measure free and total chlorine in the appropriate method range. Whenever free chlorine concentrations exceeded 0.05 mg/L, one mL of sterile sodium thiosulfate (10% *v*:*v*) was added. All probes and apparatus calibrations were performed before each sampling, according to the manufacturer’s recommendations.

#### 2.3.2. Intracellular-ATP and Flow Cytometry Assays

Fifty milliliters of water were used for intracellular-ATP quantification following the protocol specified by the manufacturer of the Dendridiag^®^ SW kit (GL-Biocontrol, Clapiers, France). Briefly, samples were filtered on 0.45 µm (Ø 33 mm) polyether sulfone (PES) sterile membranes (CLEARLine^®^ Biosigma S.p.A, Via Valletta, Italy) to eliminate free ATP and other inhibitors. Intracellular-ATP was then extracted with a solution for cell lysis provided in the kit and immediately quantified through bioluminescence assay with the Kikkoman PD-30 Luminester™ luminometer (Kikkoman Corp., Noda, Japan). At last, the measurements were validated with a standard also provided in the kit. Intracellular-ATP was expressed in picograms (pg) of ATP per millimeter and the kit had a detection limit of 0.1 pg ATP/mL.

Flow cytometry was conducted using a BD Accuri™ C6 Plus flow cytometer along with the automatic BD CSampler™ sampling arm (BD-Biosciences, Mississauga, ON, Canada) to enumerate total (TCC) and intact cell counts (ICC) based on the integrity of cell membranes as the viability criteria. Quantification of cells was performed with four small aliquots of 300 µL (per water sample) to discriminate between dead and live cells in duplicates using (1) three µL of SYBR Green fluorochrome to stain all cells, and (2) three µL of a mix of SYBR Green and propidium iodide fluorochromes to stain damaged (dead) cells. Before the addition of dyes, aliquoted samples were incubated at 37 °C for three minutes, whereas samples with dyes were incubated once again at 37 °C for 10 min in the dark. Cells were enumerated using the FL3 (>670 nm) and FL1 (530–533 nm) density plots, and bacteria gating was assessed according to the EAWAG water quality template previously developed to discriminate TCC and ICC [52]. Percentage of viable cells were calculated by dividing the number of ICC by the number of TCC.

#### 2.3.3. Liquid Culture-Based Enzymatic *Legionella pneumophila*

The quantification of culturable *L. pneumophila* through the MPN method was performed using the 100 mL potable water Legiolert/Quanti-Tray kit (IDEXX Laboratories Canada Corp., Markham, ON, Canada) according to the manufacturer’s instructions. In short, aliquots of 100 mL of water were transferred into sterile polypropylene vessels and first analyzed for water hardness using Aquadur^®^ test strips (Macherey-Nagel, Düren, Germany). Due to overall low water hardness (zero to two positive pads on test strips), 0.33 mL of Legiolert Supplement were added to each vessel, as well as one Legiolert reagent blister pack. Finally, Legiolert water sample mixtures were transferred to 96-well plates and sealed with the IDEXX Quanti-Tray Sealer PLUS. Plates were then incubated at 39 ± 0.5 °C for seven days and results were read by counting any brown or turbid wells. The MPN method ranged from 10 to 22,726 MPN/L. 

#### 2.3.4. Water Samples Filtration and DNA Extraction

Approximatively 600 to 800 mL of the remaining water sample contents were vacuum filtered on sterile 0.2 µm (Ø 47 mm) Supor^®^ PES membranes (PALL Corp., Mississauga, ON, Canada). Membranes were then gently folded and stored at −80 °C for prolonged conservation.

DNA extraction was carried out using an adapted protocol from the FastDNA Spin kit (MP Biomedicals, Solon, OH, USA). Membranes were first transferred to Lysing Matrix A tubes and fragmented with a flame-sterilized pair of scissors. A volume of 1.0 mL of a cell lysis solution for bacteria (CLS-TC) was added to each tube before homogenization in the MP Biomedicals FastPrep-24™ bead beater for two successive cycles of 40 s at 6.0 m/s. Tubes were put on ice for two minutes in between bead beating cycles. Centrifugation was then performed at 14,000× *g* for 10 min at room temperature and the supernatant (700–750 µL) was collected into sterilized polypropylene 2.0 mL microcentrifuge tubes. An equal volume of Binding Matrix to that of the collected supernatant was added, and this mixture was gently agitated at 40 rpm and room temperature on a rotator for five minutes. From that point, the instructions specified in the FastDNA Spin kit manual were identically followed. DNA extracts (100 µL) were then stored at −20 °C.

#### 2.3.5. Quantitative Polymerase Chain Reaction (qPCR) for *Legionella pneumophila*

The quantification of *L. pneumophila* DNA was conducted in triplicates by real-time qPCR according to the instructions of the iQ-Check^®^ Quanti *L. pneumophila* Real-Time PCR kit (Bio-Rad Laboratories, Mississauga, ON, Canada). Fluorescence curves were recovered on the Bio-Rad CFX Opus 96 Real-Time PCR Instrument and results were expressed in gene copies per liter. Inhibition was tested for each PCR plate in compliance with the instructions of the kit. For samples where inhibition was detected in only one of the triplicates, the inhibited triplicate was therefore not included in the analysis. Globally, amplification efficiencies and correlation coefficients (R^2^) of the qPCR standard curves ranged 94.7–96.4% and 0.992–0.996, respectively. The limit of detection was of 5 gc/reaction, whereas the mean lower and upper limits of quantification considering all PCR plates were of 19 gc/reaction and 30,285 gc/reaction. 

### 2.4. Statistical Analysis and Graphic Viewing

Data exploration and statistical analysis were conducted on Microsoft Excel version 16.59 using the Formulas tab, and graphics were sketched on Rstudio version 2021.09.0, except for one line graph which was produced with Microsoft Excel. Due to small data sets, the Student’s *t* test (“T.TEST()”) was used to compare data sets means for different purposes. Statistical significance was set at a *p*-value of 0.05. For statistical and graphic viewing purposes, samples below the detection limit for culturable *L. pneumophila* were set at 1.5 MPN/L.

## 3. Results and Discussion

### 3.1. Elevated Baseline Microbial Contamination after the 12-Week Distal Stagnation Period

After the 12-week distal stagnation period, elevated concentrations of microbial measurements (ATP, TCC, and ICC) and *L. pneumophila* in first draw showerhead samples, the water in each TMV casing, as well as the incoming hot water feed to each TMV were observed. Results are presented in Table 1 and Table 2, respectively, for the shower systems that underwent remedial flushing and shock chlorination, and serve as baseline observations to evaluate the short-term impact of these remedial interventions. Overall, concentrations were comparable to those detected at these same shower systems in a previous study after prolonged (4- or 16-week) distal water stagnation [40]. Such significant water quality losses were attributed to biofilm growth during extended stagnation and biofilm detachment occurring near water collection points.

Culturable *L. pneumophila* were detected in 83% (10/12) of first draw showerhead samples with concentrations ranging 35–6081 MPN/L, whereas *L. pneumophila* gene copies were detected in all samples at concentrations ranging 173–37,700 gc/L. Compared to levels measured prior to the start of the present study [40], a 2-log increase in culturable *L. pneumophila* was observed in one showerhead, thus showing its persistence and capacity to grow in stagnant conditions on the long-term run. Conversely, the other showerheads typically showed either a slight rebound in culturability of 0.1–0.8-log or decreases of up to 1.2-log. Reductions of 0.1–1.3-log were also measured in terms of *L. pneumophila* gene copies in most of these showerheads. Notably, one showerhead that tested negative for the presence of culturable *L. pneumophila* in the previous study remained negative at first draw in this study, despite the detection of *L. pneumophila* gene copies. These observations demonstrate that once *L. pneumophila* bacteria are established in the biofilm, it can persist over a prolonged period of time at varying concentrations during stagnation. Similarly, small increases of 1–1.6-fold in ATP and of 0.1–0.2-log in TCC and ICC were observed over the 12-week distal stagnation period. Larger increases in concentrations of ATP, TCC, ICC, and *L. pneumophila* were rapidly observed four weeks after recommissioning (remedial) flushing in the previous study [40], thus suggesting that long stagnation times (>4-week) may not systematically result in continuous microbial growth because of limiting factors in distal parts of building plumbing such as nutrients availability [33].

Although a low percentage of viable cells (14%) was measured in the water sampled from the interior of the TMV casing from one shower system, the corresponding elevated ATP and TCC concentrations of 12.79 pg ATP/mL and 8.35 × 10^6^ cell/mL were especially noteworthy (Table 1). In comparison, for the same order of magnitude of TCC in the TMV casing from the other shower system, a greater viability percentage (42%) but a lower ATP concentration (2.21 pg ATP/mL) were measured (Table 2). Such a difference could be attributable to the presence of intensive ATP intake microbial species contributing to a greater recovery of ATP molecules in the first TMV [53] despite prolific conditions of growth characterized by a higher percentage of ICC in the second TMV.

### 3.2. Microbial Concentrations from the System Are Amplified in the Distal Sections

The building incoming cold water from the municipal distribution system was characterized by low microbial concentrations (0.05 pg ATP/mL and 6.30 × 10^2^ cell/mL as of TCC) and non-detectable culturable *L. pneumophila* and gene copies, and the presence of an elevated free chlorine residual (0.72 mg/L). In the hot water return loop, ATP and TCC greatly exceeded levels found in the incoming cold water, reaching, respectively, 5.51 pg ATP/mL and 4.09 × 10^5^ cell/mL, and *L. pneumophila* was detected at culturable and qPCR concentrations of 65 MPN/L and 233 gc/L. 

The first step of microbial amplification occurred between the incoming municipal water and the hot and cold water piping feeding both TMV. Although free chlorine concentrations in cold water samples ranged from 0.27 to 0.63 mg/L throughout the study, ATP and TCC concentrations in these cold water feeds were higher by 2–17-fold and 0.3–1.3-log, respectively, in comparison to the incoming municipal cold water. Viability percentages showed wide variations (8–53%), and *L. pneumophila* was never detected in any cold water samples (Table 1 and Table 2), despite previous studies reporting frequent detection in flushed (10–15 s) cold water taps using qPCR and larger water volumes [54,55]. In contrast, the microbial amplification from the incoming municipal water was more excessive in the hot water supplied to each TMV, thus highlighting the shift in terms of conditions of growth when water temperatures increase and free chlorine residuals are depleted (0–0.08 mg/L). In fact, ATP and TCC concentrations in hot water feeds were overall higher by 12–110-fold and by 2.5-log when compared to the cold water entering the building. Culturable *L. pneumophila* was further detected at low concentrations (non-detectable to 35 MPN/L), and once at 1198 MPN/L in the hot water that remained stagnant for 12 weeks. *L. pneumophila* gene copies were for the most part in similar ranges to those detected in the hot water return loop. Then, mitigated water collected from the interior of each TMV casing showed ATP, TCC, and *L. pneumophila* gene copies that were typically in between measurements from the cold and hot water feeds, with the exception of the 12-week distal stagnation period after which mitigated water had unusually elevated microbial measurements. Therefore, stagnant water in each TMV casing was found to be utterly conducive to microbial growth, although microbial loads were further reduced with water usage during the study. 

The second step of microbial amplification occurred between the TMV and the showerheads. At the showerhead level, mean values in first draws were systematically 1.7–159-fold higher in ATP and up to 1.2-log higher in TCC than measurements from the mitigated water contained within the TMV casings despite the completion of remedial treatments and the implementation of different flushing regimes (Table 1 and Table 2). These increases in first draws showed important distal microbial amplification occurring within the showerheads and its immediate connecting pipes. Distal amplification is generally observed within the first few liters of water collected from taps [40,56,57,58] due to the high surface-to-volume ratio, the presence of heterogeneous materials and architectures, variable stagnation times, and favorable temperatures, which are all factors favoring biofilm growth [59]. *L. pneumophila* gene copies were higher by one order of magnitude in distal parts of both shower systems than the concentrations measured in the cold and hot water TMV feeds, or the mitigated water found in the TMV casings. Although flushing remains a temporary beneficial mitigation strategy to reduce microbial risks for users [40,41], flushing these showerheads for five minutes with mitigated water further reduced microbial concentrations to values generally lower than the hot water supplied to each TMV (Table 1 and Table 2). 

During this study, microbial concentrations were amplified from the incoming water to the showerheads, and the elevated microbial concentrations in the recirculated hot water appeared to be the main source of microbial cells and *L. pneumophila*. The amplification in the mitigated shower system downflow of both TMV was attributed to operational considerations, extended stagnation, and the presence of large (300 L) mitigated (22–38 °C) water networks previously identified in this building [40]. Unless efficient engineering controls are applied to the hot water distribution system, contaminated hot water can continue to seed distal points of use where favorable conditions for biofilm growth prevail. Distal amplification observed in first draws at showerheads confirms microbial regrowth, whereas flushed samples collected from these same showerheads were indicative of influent water quality. As suggested by Ji and colleagues, a deeper analysis of the building plumbing microbiome could help better understand to what extent the water eventually delivered to users is shaped by the upstream water quality and microbiome [60], so that effective controls and suitable design can thus be implemented. 

### 3.3. Impact of the Combination of Remedial Interventions with Different Flushing Regimes

#### 3.3.1. Stagnation Tends to Promote Microbial Regrowth after Remedial Intervention

Microbial regrowth factors (RFs) in the duplicates of showerheads that were left stagnant following remedial interventions are reported for first draws in Table 3 to assess whether a short stagnation period of three weeks was sufficient to promote complete regrowth from baseline data in distal sections (i.e., in that case, the factor would be of at least one). 

In the shower system that underwent remedial flushing, a 3-week stagnation period resulted in increased ATP concentrations (RF > 1) (Figure 2B) in the duplicate of stagnant showerheads, but was not long enough to fully recover TCC (RF < 1) (Figure 2C). The percentage of viable cells remained steady at 23–29% throughout the stagnation period (Figure 2D). However, complete regrowth of *L. pneumophila* culturability (RF = 1) (Figure 3A) and gene copies (RF = 1.53) (Figure 3B) was observed in one of the duplicates of showerheads, unlike the other in which only a large increase in gene copies (RF = 5.49) was measured despite culturable *L. pneumophila* cells being non-detectable. Since qPCR assays do not differentiate between culturable, viable-but-non-culturable (VBNC), and dead cells, this apparent increase of 5.49-fold (0.7-log) in *L. pneumophila* gene copies could be attributed to the detachment of a biofilm fragment or to the incoming flow of VBNC or dead cells from the upstream water when flushing was carried out. Considering that *L. pneumophila* gene copies were only detected at low concentrations (194 gc/L) in the 5 min flushed water supplied to that specific showerhead during remedial flushing, this last hypothesis could not support the 0.7-log increase. This example illustrates how two neighboring showerheads part of one grouped shower system can result in variable outcomes despite receiving the same water. Therefore, the selection of water sampling points during routine monitoring or in the aftermath of Legionnaires’ diseases cases can greatly influence environmental investigation results.

Shock chlorination followed by a 3-week distal stagnation period resulted in more important water quality losses than what was observed for the other duplicate of showerheads that underwent remedial flushing. Indeed, higher RFs were calculated for ATP (4.31 and 7.07), TCC (5.68 and 3.51), and ICC (7.70 and 4.74) in this shower system despite fairly similar baseline results (Table 3). Such a noteworthy difference among duplicates of showerheads is likely not to be the sole effect of biofilm detachment occurring after longer periods of stagnation [40], but also the result of microbial regrowth. Free chlorine typically disrupts cell membranes of microorganisms, causing leakage of macromolecules (e.g., carbon, nitrates, phosphates) and resulting in the sudden bioavailability of nutrients essential for microbial growth. Previous studies have demonstrated that some microorganisms including *L. pneumophila* [61] and mixed drinking water bacterial communities [62] can sustain necrotrophic growth. Therefore, shock chlorination combined with a 3-week distal stagnation period prompted biomass regrowth as reflected by important increases in ATP (Figure 2F), TCC (Figure 2G), and percentage of intact cells (Figure 2H) in showerheads. In contrast, this combination was not sufficient to renew *L. pneumophila* culturability and gene copies within three weeks, as RFs remained low in the duplicate of showerheads (culturable: RF = 0.02 and 0.01; qPCR: RF = 0.17 and 0.21) (Table 3). Nonetheless, stagnation did promote the resurgence of culturability (20–60 MPN/L) (Figure 3C) in both showerheads that were left stagnant, although gene copies remained near 1-log lower than baseline values (Figure 3D). Shock chlorination combined with distal water stagnation further reduced to a greater extent the ratio of culturable cells to gene copies of *L. pneumophila* than remedial flushing and stagnation, thus demonstrating its benefits on all types of viable or VBNC cells. However, *L. pneumophila* growth is likely to recur in the long-term run as the persistence of *Legionella* in building water systems has been demonstrated in several studies despite chlorination treatments [10,11,12]. Since ATP and TCC are general microbial measurements used to assess biostability [63], whereas qPCR and culturable *L. pneumophila* monitoring are pathogen-specific, it becomes essential to ensure that the microbiome did not shift towards an increased abundance of other opportunistic drinking water pathogens after such remedial treatment.

Overall, different trends were observed when distinct remedial interventions were followed by short (3-week) distal water stagnation: complete distal regrowth of *L. pneumophila*, ATP, TCC, and ICC after remedial flushing because of the limited effect of flushing on biofilm removal [64], and important amplification of ATP, TCC and ICC after shock chlorination, likely due to the release of nutrients. Longer stagnation times may however alter these observations as *L. pneumophila* concentrations can gradually increase back after shock chlorination, whereas cell counts may reach plateaued concentrations over time [64] as nutrients are depleted. Nevertheless, distal water stagnation periods of three weeks promoted microbial regrowth in all showerheads regardless of the remedial intervention, thus highlighting the need for additional preventative measures (e.g., daily or weekly flushes) when plumbing systems are to be closed or left unused following corrective actions. 

#### 3.3.2. Daily Flushes Resulted in Significantly Lower ATP and TCC Concentrations than Weekly Flushes

In this study, daily and weekly 5 min flushes of duplicates of showerheads were implemented after the completion of both remedial interventions. Such routine flushing protocols are typically recommended in several *Legionella* control guidance documents when taps are infrequently used [5,22,23,24,25,26,27,28,29,30]. Throughout the study, daily flushes resulted in significantly (*p* < 0.05) lower ATP (Figure 2B,F) and TCC concentrations (Figure 2C,G) than weekly flushes, regardless of the remedial intervention carried out. Nonetheless, the benefits of preventative flushing after shock chlorination were particularly meaningful as ATP and TCC remained lower by 5.1–16.4-fold and 0.2–0.8-log in the showerheads that were flushed either on a daily or weekly basis compared to those which were left stagnant. Therefore, periodic flushes of showerheads prevented biomass regrowth observed with the combination of shock chlorination and stagnation through the flushing of accumulated nutrients or dead microorganisms. This demonstrates the importance of maintaining a regular water flow in distal sections of plumbing systems following such remedial intervention. Whereas daily and weekly flushes slightly increased free and total chlorine residuals at first draw in showerheads (Figure 2A,E), statistical differences between those two flushing regimes were not found to be significant (*p* > 0.05). As showerheads were flushed with mitigated water, chlorine concentrations additionally remained consistently low in 5 min flushed water samples (less than 0.13 mg/L as of free chlorine and 0.33 mg/L as of total chlorine) and even more so at first draw (less than 0.01 mg/L as of free chlorine and 0.05 mg/L as of total chlorine).

Frequent flushes can prevent bacterial accumulation resulting from biofilm detachment and suspended biomass growth, therefore resulting in lower microbial concentrations such as those observed in the duplicates of daily flushed showerheads in the present study. However, when water carries a certain load of nutrients, more frequent flushes can translate into higher nutrient delivery to distal parts and in a greater potential for microbial growth [33,42], although it could be offset by the delivery of more frequent disinfectant residual if carried out with disinfected cold water. Ji and colleagues demonstrated that higher water usage frequencies led to a lower proportion of shared operational taxonomic units (OTU) between water samples and their biofilm counterparts. This observation was attributed to the fact that microbial interactions among these two phases were less likely to occur during intermittent and short stagnation times [65]. Consequently, when regular flushing is carried out under suitable preventative control regimes in terms of temperature or disinfectant residuals, the user is more likely to be exposed to upstream water that is typically characterized by lower microbial concentrations than the first few liters of water in distal sections, thus reducing microbial risks.

In this study, all water samples were collected on Mondays, thereupon after weekend stagnation to accommodate building staff availability. However, previous studies have shown that weekend-long stagnation periods can increase TCC in first draws by less than 1-log compared to measurements taken right before the start of the weekend [57,58,66]. Indeed, the detachment in faucets was observed to occur mostly over the first 24 h of stagnation [58]. Therefore, it is likely that statistical differences between daily and weekly flushes could be even greater if samples had not been collected after the weekend. Future research should aim to integrate daily flushes with cold, mitigated, or hot water in auto-flush devices to prevent water stagnation and microbial growth. 

#### 3.3.3. The Combination of Preventative Flushing and Shock Chlorination Is the Most Effective to Reduce Temporarily *L. pneumophila*

Overall, larger decreases in both *L. pneumophila* culturability (Figure 3C) and gene copies (Figure 3D) were observed during the study after shock chlorination comparatively to remedial flushing (Figure 3A,B). No culturable *L. pneumophila* cells were measured in first draws over the first and second week after shock chlorination, regardless of the flushing regime implemented and despite flushes supplying up to 834 gc/L (five min flushed samples) each time, whereas two showerheads persistently showed concentrations ranging 11–223 MPN/L after remedial flushing. *L. pneumophila* gene copies at first draw generally remained 1-log lower than baseline values in showerheads that underwent shock chlorination, whilst concentrations persisted in the same orders of magnitude (10^2^–10^4^ gc/L) than baseline measurements following remedial flushing. In general, the ratio of culturable to qPCR concentrations was diminished to a greater degree following shock chlorination than after remedial flushing, but there were no notable differences in these ratios when considering the flushing regime then implemented in each shower system. As the discrepancy between culturable and qPCR *L. pneumophila* concentrations typically provides insights into the presence of VBNC and dead cells, the combination of preventative flushing and shock chlorination was more effective in reducing the proportion of VBNC and dead cells in this study. 

Despite daily flushed showerheads being repeatedly exposed to temperatures favorable for *L. pneumophila* growth (32–40 °C) for 5 to 55 min after each flush (Figure 4), daily flushes did not yield statistically less *L. pneumophila* at first draw than weekly flushes (*p* = 0.15–0.57). The benefits of regularly washing cells away with the flow were therefore more important than the temporary establishment of conditions optimal for *L. pneumophila* growth in distal parts [67]. Nonetheless, the combination of remedial flushing or shock chlorination and daily flushes resulted in more important decreases than weekly flushes in terms of *L. pneumophila* culturability and gene copies over the first week after the intervention, as well as between baseline measurements and the third week of the study. By the end of the study, daily and weekly flushing of showerheads generally resulted in lower *L. pneumophila* culturable and qPCR concentrations than showerheads left stagnant, therefore demonstrating the beneficial effects of sustained (3 weeks) preventative flushing when combined with remedial interventions.

Despite the clear short-term benefits (3 weeks) of shock chlorination on *L. pneumophila*, such remedial intervention can accelerate plumbing corrosion and cause the formation of harmful disinfection by-products [20]. Additionally, it requires the assistance of professionals and prolonged time windows regarding the interruption of water usage. In this study, free chlorine residuals ranging from 21.9 to 25.2 mg/L were maintained at all showerheads for 16 h. Figure 3 shows culturable *L. pneumophila* regrowth observed by the end of the 3-week study period in some showerheads, which was likely due to (1) recolonization of *L. pneumophila* through flushing and seeding of planktonic unculturable *L. pneumophila* that could regain culturability over time, (2) persistence of *L. pneumophila* in the distal biofilm, and (3) the protection of *L. pneumophila* cells in protozoan hosts for a short period of time [15,16,17,18]. The limited short-term impact of remedial flushing on *L. pneumophila* can be attributable to the mechanical action of flushing, which only contributed to the removal of superficial biofilm cells that were poorly attached and planktonic cells controlled by the flow [68]. These general trends should also take into consideration that the baseline concentrations of culturable and qPCR *L. pneumophila* varied substantially between showerheads, even though they were engineered identically and positioned next to each other within the same grouped shower system. Interestingly, incoming concentrations of culturable and qPCR *L. pneumophila* after flushing that renew cells to distal points remained quite stable over time, reflecting the steady operations of the upstream distribution system.

#### 3.3.4. Daily Thorough Flushes of Showerheads for Months Can Reduce the Occurrence of Culturable *L. pneumophila* in Distal Sections

Even if providing actionable information to restart a system after extended stagnation, reduced building occupancy, or contamination events, this study does not provide more long-term evidence to assess whether the trends observed after both remedial interventions persist over time as the plumbing microbiome matures after being temporarily shifted. Long-term data would be beneficial to estimate the time duration by which *L. pneumophila* concentrations are likely to return to baseline values or not. Nonetheless, results presented in this study represent valuable information for building managers who want to temporarily lower microbial risks while implementing additional engineering controls (e.g., temperature correction) or building-wide treatment (e.g., in situ chloramine system). In contrast to distal water stagnation, the benefits of preventative flushing, and more particularly daily flushes of showerheads, combined with remedial interventions, hereby remedial flushing, and shock chlorination, were well demonstrated on the concentrations of ATP, TCC, and the percentage of intact cells, and to a lesser extent on the abundance of *L. pneumophila*. 

As the pandemic persisted, building managers faced the issue of temporary closings and underuse of the facilities when reopening. To mitigate the potential risk to users, building managers thus implemented manual daily flushes of showerheads in most shower systems from mid-September 2021 to early January 2022 by flushing the rear-end showerhead of large, grouped shower systems for 15 to 30 min, followed by a brief 30 s flush of all upstream showerheads. In fact, this approach has been considered as a more time-efficient alternative in recommissioning guidance [38] to minimize the duration of flushing operations and the wastage of water after prolonged building inoccupation. Two months after implementing this modified flushing regime, *L. pneumophila* culturability persisted at a positivity rate of 41% (11/27) in first draws despite prior daily flushes, with positive concentrations ranging from 10 to more than 22,726 MPN/L (mean of positive samples of 3163 MPN/L). It is noteworthy that all (n = 4) rear-end showerheads in each selected large, grouped shower system remained below the detection limit of 10 MPN/L for culturable *L. pneumophila*, with the exception of one showerhead in which a low concentration of 74 MPN/L was measured. These levels were more than 3-log lower than those measured before the implementation of daily 15–30 min flushes (mean of 3422 MPN/L). This shows that in a large, grouped shower system, a brief daily flush of 30 s as opposed to more thorough flushes of rear-end water points as a time-efficient flushing strategy was not sufficient to depress culturable *L. pneumophila* in distal sections. Nonetheless, long-term (2-month) daily flushing of showerheads resulted in reduced culturable *L. pneumophila* concentrations below desirable thresholds of 1000 and 10,000 colony-forming unit (CFU)/L (i.e., near equivalent of MPN/L [69,70,71]) in, respectively, 24 and 26 out of the 27 investigated showerheads.

## 4. Conclusions and Recommendations

In this study, the weekly short-term (3-week) effectiveness of the combination of shock chlorination (20–25 mg/L, 16 h) or remedial flushing (five minutes at each showerhead) with different flushing regimes (daily, weekly, left stagnant) were implemented in duplicates of showerheads. Overall, this work demonstrated the temporary benefits of carrying remedial interventions despite the application of preventative flushing regimes. 

The following recommendations are proposed to support building managers and other relevant authorities in the development of effective water management plans and environmental monitoring strategies.
In general:
**Monitoring microbial concentrations in plumbing piping sections from the incoming water to the points of use can locate sites of contamination so that targeted engineering controls or in situ treatments can be effectively applied.** In this study, ATP, TCC, and *L. pneumophila* concentrations increased from the building cold water entry to the hot water return loop, then from the cold and hot water supplied to each shower systems’ TMV towards each showerhead. Amplification of ATP, TCC, ICC, and *L. pneumophila* occurred clearly at distal sites as opposed to the upstream distribution system.**System-specific *L. pneumophila* alert and action thresholds should be set to ensure minimum amplification at distal sites using qPCR as the first-tier surveillance and culture as the confirmation.** Throughout this study, qPCR was a more conservative mean to assess the weekly effectiveness of remedial interventions as it provided fairly steady measurements comparatively to culturable measurements which varied more substantially.**In grouped distal sites, more than two water points should be at least monitored during routine (baseline) or investigative *L. pneumophila* monitoring.** Neighboring showerheads that were engineered identically and received the same building plumbing water showed fairly different *L. pneumophila* culturable and qPCR concentrations in baseline measurements and had variable response to remedial interventions carried out.
For buildings with established contamination at distal sites:
**A combination of shock chlorination/preventative flushing is more effective than remedial flushing/preventative flushing to control temporarily (3-week) the regrowth of *L. pneumophila* at distal sites.** Throughout the study, such combination (shock chlorination/preventative flushing) led to a 2-week suppression of *L. pneumophila* culturability and to greater decreases in qPCR concentrations than remedial flushing/preventative flushing for which *L. pneumophila* persisted at 11–223 MPN/L and 102–104 gc/L.**The combination of preventative flushing (daily or weekly) and shock chlorination should be considered as a temporary measure to limit growth of *L. pneumophila*** as it provided protection for at least three weeks at distal sites before small rebounds in culturability (20–84 MPN/L) were observed. Alternative mitigation strategies and engineering controls should consequently be considered to limit its long-term regrowth.**Shock chlorination should not be followed by long periods (more than 3 weeks) of water stagnation as stagnant conditions can stimulate biomass regrowth. Such remedial intervention should therefore not be systematically conducted prior to low building occupancy or complete long-term building shutdown unless preventative flushing is implemented to wash away dead cells and nutrients to prevent microbial growth and necrotrophic growth.** Indeed, larger microbial regrowth factors (RFs) were measured in showerheads left stagnant for three weeks after shock chlorination than following remedial flushing for concentrations of ATP (shock chlorination: RFs = 4.31 and 7.07; remedial flushing: RFs = 1.48 and 1.18), TCC (shock chlorination: RFs = 5.68 and 3.51; remedial flushing: RFs = 0.86 and 0.82) and ICC (shock chlorination: RFs = 4.74 and 7.70; remedial flushing: RFs = 0.74 and 0.83).
For buildings subjected to periods of low water use due to reduced occupancy (e.g., seasonal venues, partial building shutdown, construction activities):
**Without further evidence, daily flushes of distal sites as a measure to prevent water stagnation hazards should be considered where occupant exposure (e.g., showerheads) and susceptibility (e.g., vulnerable or immunocompromised individuals) are more risk critical.** Indeed, regardless of the remedial intervention carried out (remedial flushing or shock chlorination), daily flushes of showerheads resulted in significantly (*p* < 0.05) lower ATP and TCC concentrations, as well as in generally lower *L. pneumophila* levels in this study.



## Figures and Tables

**Figure 1 microorganisms-11-01361-f001:**
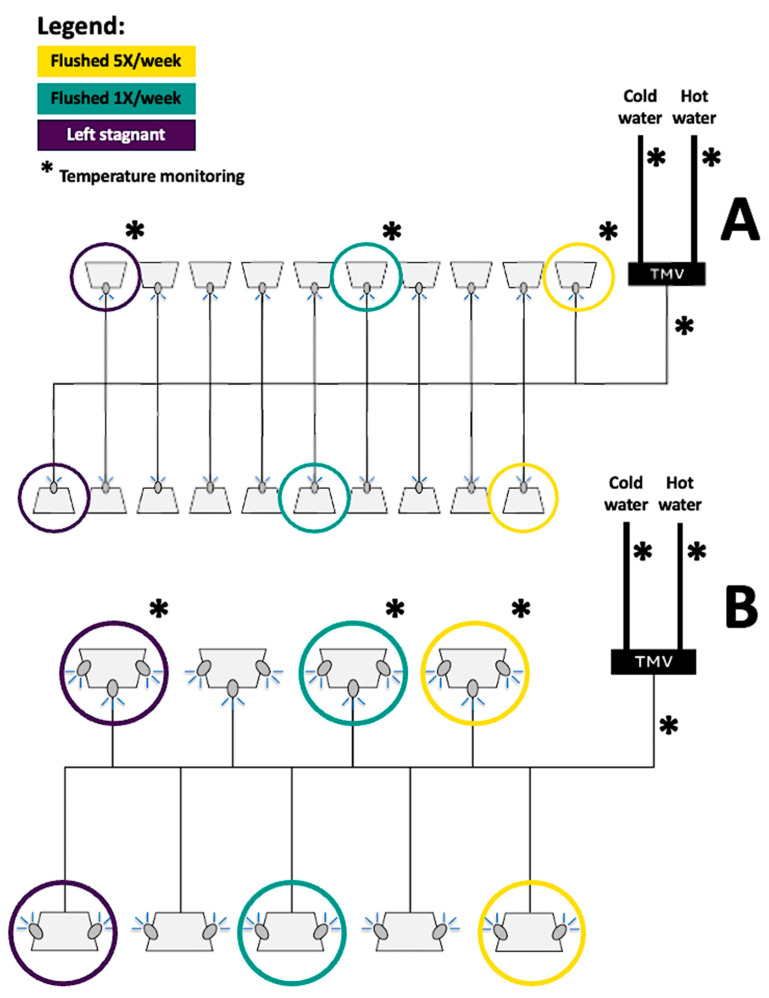
Schematic of the interventions and temperature monitoring performed in the shower systems in which (**A**) remedial flushing and (**B**) shock chlorination were, respectively, carried out.

**Figure 2 microorganisms-11-01361-f002:**
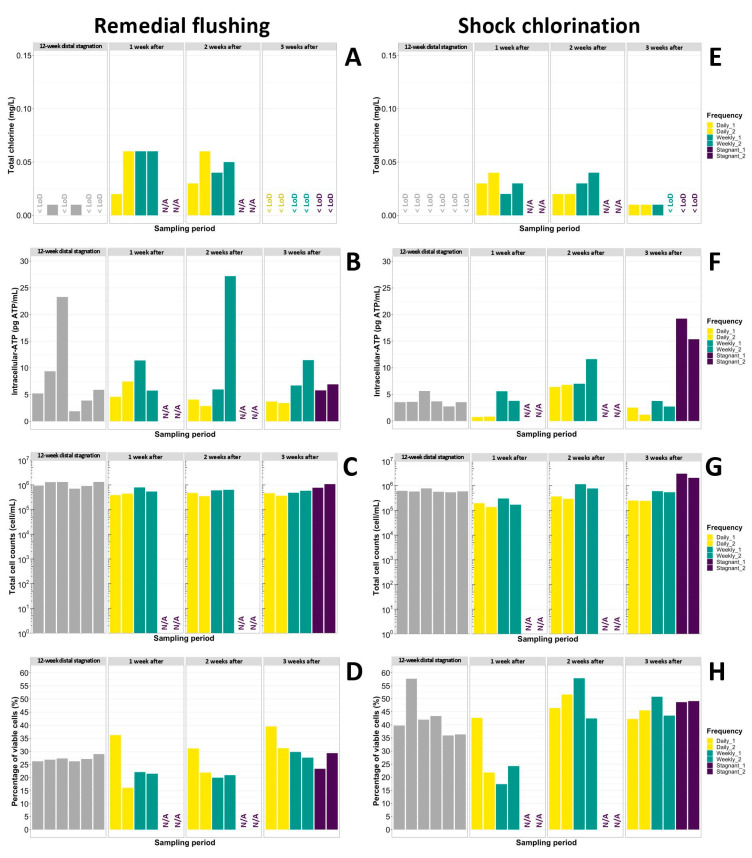
Weekly bar plot representations of (**A**) Total chlorine, (**B**) Intracellular-ATP, (**C**) Total cell counts, and (**D**) Percentage of viable cells after remedial flushing was carried out in one shower system, and weekly bar plot representations of (**E**) Total chlorine, (**F**) Intracellular-ATP, (**G**) Total cell counts, and (**H**) Percentage of viable cells after shock chlorination was carried out in the second shower system. Grey bars—Baseline (12-week distal stagnation) values; N/A—Not evaluated; <LoD—Below the detection limit.

**Figure 3 microorganisms-11-01361-f003:**
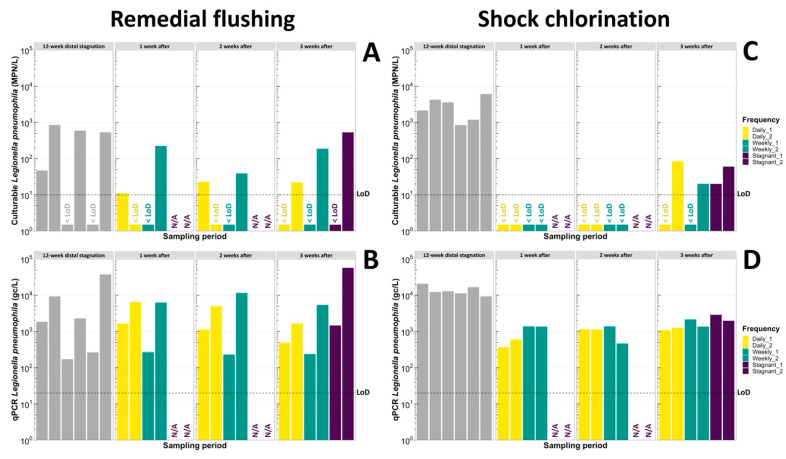
Weekly bar plot representations of (**A**) Culturable *L. pneumophila* and (**B**) qPCR *L. pneumophila* after remedial flushing was carried out in one shower system, and weekly bar plot representations of (**C**) Culturable *L. pneumophila* and (**D**) qPCR *L. pneumophila* after shock chlorination was carried out in the second shower system. Grey bars—Baseline (12-week distal stagnation) values; N/A—Not evaluated; <LoD—Below the detection limit.

**Figure 4 microorganisms-11-01361-f004:**
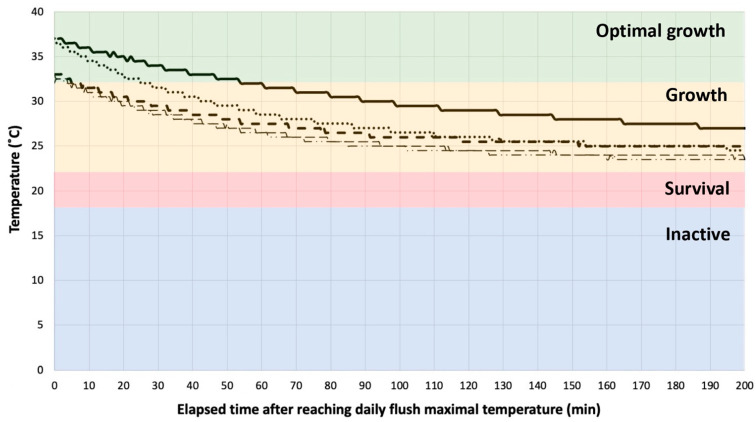
Temperature (*y*-axis) and elapsed time after reaching daily flush maximal temperature (*x*-axis) in showerheads flushed on a daily basis. Lines represent five different sets of temperature monitoring data over time. Colored temperature ranges—Typical conditions of growth of *Legionella pneumophila*.

**Table 1 microorganisms-11-01361-t001:** Results of microbiological measurements in the shower system in which remedial flushing was carried out after the 12-week distal stagnation period (n = 1 for each TMV sample; n = 4–6 samples for distal and system samples). Legend: CW—Cold water, HW—Hot water, MW—Mitigated water, *Lp*—*Legionella pneumophila* concentrations, <LoD—Below detection limit.

Sampling Period	Water Sample	ATP (pg ATP/mL)	TCC (cell/mL)	Percent Viability (%)	Culturable *Lp* (MPN/L)	qPCR *Lp* (gc/L)
12-week distal stagnation	TMV—CW	0.84	7.10 × 10^3^	8	<LoD	<LoD
	TMV—HW	0.80	1.45 × 10^5^	22	<LoD	622
	TMV—MW	12.79	8.35 × 10^6^	14	<LoD	427
	Distal—1st draw (mean)	8.26	1.07 × 10^6^	27	340	8603
	System—5-min (mean)	0.36	1.40 × 10^5^	5	<LoD	1087
1 week after remedial flushing	TMV—CW	0.08	1.87 × 10^3^	53	<LoD	<LoD
	TMV—HW	1.28	2.30 × 10^5^	24	23	415
	TMV—MW	4.21	3.69 × 10^5^	38	<LoD	419
	Distal—1st draw (mean)	7.30	5.40 × 10^5^	24	61	3657
	System—5-min (mean)	0.29	1.23 × 10^5^	7	<LoD	248
2 weeks after remedial flushing	TMV—CW	0.04	1.82 × 10^3^	43	<LoD	<LoD
	TMV—HW	1.88	3.04 × 10^5^	24	35	394
	TMV—MW	1.78	2.80 × 10^5^	36	<LoD	554
	Distal—1st draw (mean)	10.02	5.15 × 10^5^	24	18	4460
	System—5-min (mean)	0.13	9.07 × 10^4^	7	<LoD	189
3 weeks after remedial flushing	TMV—CW	0.08	3.39 × 10^3^	35	<LoD	<LoD
	TMV—HW	1.10	1.50 × 10^5^	20	<LoD	235
	TMV—MW	2.40	2.34 × 10^5^	33	<LoD	149
	Distal—1st draw (mean)	6.33	6.22 × 10^5^	30	126	11,118
	System—5-min (mean)	0.15	1.23 × 10^5^	4	<LoD	531

**Table 2 microorganisms-11-01361-t002:** Results of microbiological measurements in the shower system in which shock chlorination was carried out after the 12-week distal stagnation period (n = 1 for each TMV sample; n = 4–6 samples for distal and system samples). Legend: CW—Cold water, HW—Hot water, MW—Mitigated water, *Lp*—*Legionella pneumophila* concentrations, <LoD—Below detection limit.

Sampling Period	Water Sample	ATP (pg ATP/mL)	TCC (cell/mL)	Percent Viability (%)	Culturable *Lp* (MPN/L)	qPCR *Lp* (gc/L)
12-week distal stagnation	TMV—CW	0.27	1.26 × 10^4^	12	<LoD	<LoD
	TMV—HW	0.62	1.45 × 10^5^	20	1198	541
	TMV—MW	2.21	2.57 × 10^6^	42	3071	44,000
	Distal—1st draw (mean)	3.79	6.15 × 10^5^	43	3017	13,900
	System—5-min (mean)	0.35	1.63 × 10^5^	11	10	6405
1 week after shock chlorination	TMV—CW	0.29	8.60 × 10^2^	15	<LoD	<LoD
	TMV—HW	1.03	1.41 × 10^5^	10	11	428
	TMV—MW	0.28	6.36 × 10^4^	1	<LoD	1480
	Distal—1st draw (mean)	2.74	2.01 × 10^5^	27	<LoD	920
	System—5-min (mean)	0.27	9.64 × 10^4^	3	<LoD	441
2 weeks after shock chlorination	TMV—CW	0.01	9.60 × 10^2^	47	<LoD	<LoD
	TMV—HW	0.96	1.53 × 10^5^	22	10	385
	TMV—MW	0.05	6.25 × 10^4^	6	<LoD	860
	Distal—1st draw (mean)	7.96	6.48 × 10^5^	50	<LoD	1026
	System—5-min (mean)	0.12	1.20 × 10^5^	4	<LoD	398
3 weeks after shock chlorination	TMV—CW	0.03	1.56 × 10^3^	26	<LoD	<LoD
	TMV—HW	0.83	1.73 × 10^5^	16	<LoD	700
	TMV—MW	0.12	7.38 × 10^4^	4	<LoD	2990
	Distal—1st draw (mean)	7.45	1.13 × 10^6^	47	32	1783
	System—5-min (mean)	0.50	1.45 × 10^5^	6	ND	858

**Table 3 microorganisms-11-01361-t003:** Microbial regrowth factors in duplicates of showerheads that were left stagnant after remedial interventions (RF = value at the third week after the intervention over baseline value).

Intervention	Shower ID	ATP (pg ATP/mL)	TCC (cell/mL)	ICC (cell/mL)	Culturable *Lp* (MPN/L)	qPCR *Lp* (gc/L)
Remedial flushing	Stagnant_1	1.48	0.86	0.74	0	5.49
	Stagnant_2	1.18	0.82	0.83	1.00	1.53
Shock chlorination	Stagnant_1	7.07	5.68	7.70	0.02	0.17
	Stagnant_2	4.31	3.51	4.74	0.01	0.21

## Data Availability

Not applicable.

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
