# Peer review of "Controlling Legionella pneumophila in Showerheads: Combination of Remedial Intervention and Preventative Flushing"

_microorganisms, 2023, doi:10.3390/microorganisms11061361_

Round 1

Reviewer 1 Report

The paper shows an experimental test performed on a shower system to assess the impact of  remedial intervention and preventative flushing on Legionella spreading. The topic is worth of investigation. The paper is well-written and clear. The methodology is robust, the results are properly discussed.

I  only have one suggestion before publication: to add at least two figure to visualize i) the P&ID of the testing system (including temperature sensors location), and ii) the schematic of the test performed, in order to help the readers in rapidly and cleary understand these aspects.

Reviewer 2 Report

Grimard-Conea et al. reported and researched methods to control Legionella pneumophila in showerheads. The article is well constructed, and the results and discussion are abundant. The following lists indicate my concerns.

Line 22: The full name of the abbreviation “MPN” should be shown for the first time.

Line 30: The author described Legionella pneumophila as an opportunistic drinking water pathogen. In my opinion, LP, of course, is a pathogen that is widely distributed in aquatic environments, including water and moist soils. However, thinking of it as a drinking water pathogen is inappropriate, for its pathogenic process is not related to drinking, but is inhaling aerosols containing LP. I advise the author to express it as “Legionella pneumophila is an opportunistic pathogen that can be distributed in drinking water systems...

Line 234: the method section, the author used intracellular-ATP and flow cytometry assays to determine the water pollution or the cells in the water. I did not find these two tests to be LP-specific, because it seems that the pollution is not necessarily caused by LP or that the cells were all LP (e.g., it could be other bacteria). But the topic of the paper focused on LP in showerheads. I think the author should carefully consider this issue and make some improvements in the wording. The authors can combine the data of TCC, ICC, MPN, and gene copies of LP per liter to research their correlation relationships, it may provide more detailed information on the effect of controlling (e.g., remedial intervention and preventative flushing).

Line 235: “Fifty millimeters of water were used for intracellular-ATP quantification...” Why is a unit of length used here to quantify water?
